# The Southern Ocean marine ice record of the early historical, circum-Antarctic voyages of Cook and Bellingshausen

Grant R. Bigg[1]

[1]Department of Geography, University of Sheffield, Sheffield, S10 2TN, U.K.

5 *Correspondence to*: Grant R. Bigg (grant.bigg@sheffield.ac.uk)

**Abstract.** The circum-navigations of Cook's Second Voyage (1772-1775) and Bellingshausen (1819-1821) were attempts to find any great southern land mass poleward of ~50ºS and consequently involved sailing for three or two summers respectively in polar latitudes around Antarctica. Extensive sea ice eventually blocked each voyages' southern probes, although Bellingshausen, unknowingly at the time, saw the Antarctic continent. However, these attempts meant sea-ice and 10 iceberg records from the early historical period were collected near simultaneously from around much of Antarctica. Here, these records are extracted from journals, analysed, and compared to each other and the modern satellite record of both forms of marine ice. They generally show an early historical period with a more northerly record of both forms of marine ice than normal for today, but to a geographically varying degree. However, the early historical period in the Pacific sector of the Southern Ocean saw marine ice generally within the range of modern observations for the same time of year, but the 15 Weddell Sea and Indian Ocean marine ice, particularly on Cook's voyage, then extended several degrees further north than in today's extreme ice years.

## 1 Introduction

Marine ice, whether sea ice or icebergs, as well as extensive land, had long been realised to be an effective barrier to northward maritime travel in the North Atlantic (Goodwin, 2019) and, by the mid-eighteenth century, in the northern seas of 20 the Pacific (McCannon, 2012). However, while isolated reports of icebergs occurred from at least 1687 in the southwest Atlantic and Drake Passage (Martin et al., 2022; Headland et al., 2023) far less was known about the Southern Ocean by the 1770s. Ships rounded Cape Horn to travel between the Atlantic and Pacific Oceans, but while such voyages were frequently stormy all but a handful of unlucky voyagers stayed close enough to South America to miss encounters with icebergs. The search for Terra Australis, while leading to European discovery of South Pacific islands, as well as New Zealand and 25 Australia, had not extended south of temperate latitudes away from South America.

The British Admiralty and the Royal Society jointly sponsored James Cook's second expedition, starting in 1772, to search for land in southern latitudes. This followed his first global circum-navigation expedition (1768-1771) where he had charted both islands of New Zealand, and travelled up much of the east coast of Australia, reducing the possible existence of a southern land mass to, at best, sub-polar southern latitudes. Interestingly, in rounding Cape Horn during that voyage on the

way to observing the transit of Venus from Tahiti, his journal records no encounter with marine ice even though the *Endeavour* reached 60ºS (Cook, 1771). The Admiralty also sent an expedition towards the North Pole the following year, under the command of Constantine Phipps, famous for the presence of Midshipman Horatio Nelson (Goodwin, 2019). However, there does not appear to have been any overt link between these two contemporary British polar expeditions.

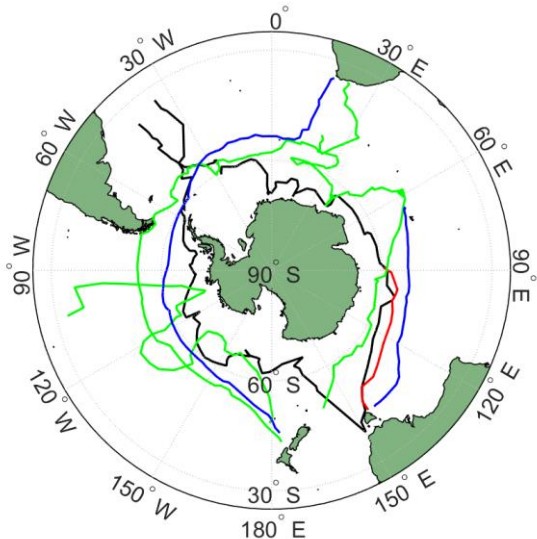

**Figure 1: Southern Ocean sections of the voyages considered in this paper; green line: *Resolution*, blue line: *Adventure*, black line: *Vostok*, red line: *Mirny*. Note that Cook's outward South Atlantic section is not shown until he encountered ice (~ 50.7ºS, 20.3ºE), while his return is shown all the way to Cape Town.**

Cook, in the *Resolution*, together with Tobias Furneaux, in the *Adventure*, headed south from Cape Town in November 1772 and over three austral summers they made several attempts to reach as far south as possible before sea ice became extensive enough, or icebergs became too frequent, to allow further southing (Fig. 1). The expedition first managed to cross the Antarctic Circle in January 1773, near 40ºE, before being turned back by extensive pack ice. The two vessels became separated in poor weather in early February 1773, near 50ºS, 64ºE in the southern Indian Ocean, and then made their

separate ways to a meeting point in New Zealand, for over-wintering. Furneaux, with *Adventure*, headed roughly due east, for Tasmania, to confirm that Australia did not extend deeply south. In contrast, Cook in the *Resolution*, turned south but while spending some weeks in sub-polar latitudes only reached 61ºS in this stretch because of recurring dense iceberg fields. After rendezvousing in New Zealand, both ships spent the austral winter exploring the South Pacific island belt. They were separated during a storm in October 1773 on the return voyage back to New Zealand, from where they had planned to start a

second polar leg across the Pacific. Cook reached the rendezvous point first and after waiting in vain for *Adventure* to join him, he eventually decided to set sail on 26[th] November 1773, to begin this second leg with a full summer sailing season

ahead of him. Furneaux arrived at the rendezvous four days later. Cook had left a message that he would return to Queen Charlotte Sound in New Zealand after the 1773/4 summer, but Furneaux, leaving a message of his intentions, decided to continue on a southerly course across the Pacific and South Atlantic, heading for Cape Town, and then Britain, in 1774. The

55 *Adventure* reached 61ºS around 90ºW and then 60ºS in the Drake Passage, during this return journey, sighting a number of icebergs on approaching the South Atlantic (see separate track in Fig. 1, and Table 1 for a summary of the various ships and sections covered in this paper).

**Table 1.** Details of the historical records used in this study for each separate ship's polar sections. A '+' sign in the "ship"

column means both vessels of the particular expedition were together. The sector column shows the approximate longitude range covered in the period given in the Dates column.

| Dates | Ship | Sector | Captain |
|---|---|---|---|
| 10 December 1772 - 8 February 1773 | Resolution+ | 0-63.5ºE | Cook |
| 9 February 1773 – 22 March 1773 | Resolution | 63,5-160ºE | Cook |
| 9 February 1773 – 10 March 1773 | Adventure | 64-143ºE | Furneaux |
| 2 December 1773 – 28 February 1774 | Resolution | 180-95ºW | Cook |
| 26 December 1773 – 2 February 1774 | Adventure | 175ºE-60ºW | Furneaux |
| 3 February 1774 – 16 March 1774 | Adventure | 60ºW-18ºE | Furneaux |
| 12 November 1774 – 27 December 1774 | Resolution | 175ºE-70ºW | Cook |
| 29 December 1774 – 17 March 1775 | Resolution | 70ºW-35ºE | Cook |
| | | | |
| 13 December 1819 – 16 March 1820 | Vostok+ | 43ºW-86ºE | Bellingshausen |
| 17 March 1820 – 5 April 1820 | Mirny | 88ºE-145ºE | Lazarev |
| 17 March 1820 – 7 April 1820 | Vostok | 86ºE-150ºE | Bellingshausen |
| 2 December 1820 – 3 February 1821 | Vostok+ | 159ºE-71ºW | Bellingshausen |
| 5 February 1821 – 28 February 1821 | Vostok+ | 63ºW-31ºW | Bellingshausen |

In contrast, Cook's austral summer of 1773/4 was spent searching for southing. Initially Cook headed southeast, dodging

icebergs and sea ice for some time beyond 60ºS in the South Pacific, eventually being turned back by extensive pack ice just beyond the Antarctic Circle, near 140ºW (Fig. 1). After returning to more temperate climes Cook again headed south, reaching beyond 71ºS near 107ºW, before once more being turned back by extensive iceberg fields and pack ice. With the decline of summer at the beginning of February, 1774 Cook headed north, around 100ºW, to begin his return to over-winter in New Zealand. For his final austral summer of southern exploration (Table 1), Cook first crossed the Pacific to spend

Christmas 1774 near the tip of South America and then headed southwest, discovering South Georgia before being turned back by sea ice and icebergs near 60ºS, 30ºW, in the South Atlantic and then heading for Cape Town (Fig. 1).

Cook's expedition had shown decisively that there was no great Terra Australis awaiting discovery, at least outside of high latitudes. However, he speculated on the existence of a polar land mass as being necessary to supply the large number of icebergs seen in the Southern Ocean, as he correctly believed that the glaciers seen on South Georgia could not possibly "produce the ten thousandth part of what we have seen" of icebergs (Cook, 1775). This led to the 1819-1821 Russian Antarctic expedition of Fabian Bellingshausen (Table 1) who was commissioned to extend Cook's explorations even further south in the hope of discovering the southern continent that had to be the source of the Southern Ocean icebergs. Bellingshausen, in the *Vostok*, accompanied by Mikhail Lazarev in the *Mirny*, reached South Atlantic waters in December 1819 and spent the next two austral summers in a combination of repeating, and where possible, extending southwards, Cook's path (Fig. 1). After passing South Georgia they also encountered significant sea ice and iceberg fields near 60ºS, 30ºW, as had Cook 44 years earlier. However, they managed to continue east and reached 69ºS near the Greenwich meridian on 24 January 1820 (New Style, see section 2.1 for discussion of dating), likely sighting Antarctica, or fast ice connected to the continent, several times over the next month, before they were forced to turn decisively northwards near 40ºE. They continued seeing icebergs just north of 60ºS until near 110ºE, at which point they headed for Sydney, Australia to over-winter. Note that the *Mirny* became separated from the *Vostok*, during a storm on 17th March (NS), 1820, near 60ºS, 89ºE and continued to Sydney separately (Table 1 and Fig. 1).

Next spring they returned to the Southern Ocean (Table 1), reaching ~ 65ºS by late November 1820 near 160ºE, where southward sailing was prevented by pack ice and icebergs (Fig. 1). Heading east, they again attempted to head south around 170ºW, managing to cross the Antarctic Circle to ~ 67ºS, but again were prevented from journeying further southward by more pack ice. However, they continued along the edge of the pack ice for ~ 20º of longitude, before being driven northwards by an extensive iceberg field. Continuing eastward at high latitudes, however, they attempted another southward excursion at the approach of the new year of 1821, again crossing the Antarctic Circle to exceed 67ºS near 120ºW, before again being blocked by pack ice. A week later another southward foray reached in excess of 69ºS, where they remained below the Antarctic Circle for a few days travelling eastward from ~ 95ºW-75ºW (Fig. 1). Pack ice pushed them north again, after which they sailed through the Drake Passage at a high latitude of ~ 60ºN, seeing only an occasional iceberg, before turning for Cape Town and home at 50ºW at the end of January 1821.

A number of other expeditions explored the Weddell Sea over the next few decades (see (Love and Bigg, 2023) for a summary) and others visited parts of the Antarctic coastline during the nineteenth century, such as Ross and Crozier in the Ross Sea during the 1840s (Palin, 2018) and the approach of the Challenger expedition to the Indian Ocean sector in the 1870s (Jones, 2022) there were no other near-synchronous, geographically extensive, surveys of the far Southern Ocean until the Heroic Age of Antarctic Expedition at the turn of the twentieth century (Edinburgh and Day, 2016). Even then, the Indian Ocean and central Pacific sectors of the Southern Ocean were not visited. The Cook and Bellingshausen expeditions

therefore give us two unique snapshot views of Southern Ocean marine ice cover over a hundred and fifty years prior to regular comprehensive satellite coverage. The purpose of this paper is to inter-compare these and examine their records relative to the extensive post-1978 satellite coverage of sea ice and icebergs.

There are some existing long-term iceberg and sea ice data or reconstructions with which this work can be set in context. The pioneering work of Parkinson (1990) first revealed some of Cook's record of more extensive sea ice in the Weddell Sea, while also noting evidence of Cook's and Bellingshausen's other sea ice records being within normal range. A recent study by Martin et al. (2022) has also extracted Cook's records of iceberg and sea-ice from his 1772-1775 expedition, showing that, apart from a much wider eastward expansion of sea ice in the Weddell sea ice tongue, their data fits within the envelope of modern observations; here we not only compare our independent reconstruction with their dataset but we extend this with sea ice records, and extracts of both variables from the separate journeys of Furneaux in the *Adventure*. Headland et al. (2023) produced a dataset of Southern Ocean iceberg records from 1687-1933, although none of Cook's or Bellingshausen's iceberg data are included within this dataset. With regard to Southern Ocean sea ice there are two proxy reconstructions using different approaches, one back to 1900 by Fogt et al. (2022) and another by Dalaiden et al. (2023) back to 1700. These previous works will be considered in the Discussion, but it is worth beginning the work by noting our principal hypothesis that the two circum-Antarctic expeditions considered here occurred at the height of the Little Ice Age, so it is expected that sea ice and iceberg records will generally extend further north than those today. The validity, and geographical consistency, of this hypothesis will be seen below.

## 2 Data and Methods

### 2.1 Documentary sources

The key data sources underlying this study are daily journals and logbooks from the voyages of Cook and Bellingshausen. For Cook's expedition (1772-1775) those used include the post-voyage journals of Cook (Cook, 1775) and Johann Reinhold Forster (1981), both of whom were on board the *Resolution*, and the logbook of Tobias Furneaux (1774), captain of the *Adventure*. From Forster (1981), it is Volumes II-IV that are relevant to the Southern Ocean part of the expedition. It is worth noting that, for Cook's voyage, these sources are different from those used by Martin et al. (2022), thus providing a new dataset for comparison. For Bellingshausen's voyage (1819-1821) they include a translation of the journal of Bellingshausen (Bellingshausen, 2016), where Volume II contains the Southern Ocean component. Note that the separate voyage of the *Mirny* in the Indian Ocean sector is included within this journal.

All journals and logs were read and where sea ice or icebergs were mentioned a set of data were recorded in a spreadsheet (Bigg, 2024). These entries also include days at high latitude before and after the last ice encounters. The positional data recorded were the day, month and year of the record, the latitude and (where recorded) the longitude at noon on the day. Where, very occasionally, either the latitude or longitude is not given for a day with marine ice observations a value is found by averaging neighbouring day's positions. Very occasionally, the position was given at a different time of the day,

presumably through lunar rather than solar observations, but for the purposes of this study this time difference was ignored. Bellingshausen used the Russian "Old Style" calendar, so his observations are 12 days earlier than dates given by the contemporary calendar; in the spreadsheet all his data has therefore been adjusted forward 12 days for consistency.

Cook and Furneaux's voyage took place early in the age of using chronometers to determine longitude (Sobel, 1996). Both captains had a copy of Harrison's K2 chronometer on board their respective ships for time-keeping relative to known meridians. These chronometers gradually lost time so during sections of their voyages with no sight of known land longitude values derived purely from the chronometer accumulated error. Cook and Forster had corrected this on return to Britain, to standardize the daily longitude measurements in their journals. However, Furneaux's log records longitude as given by the time difference between the chronometer and observed noon. The chronometer was re-set at the known positions of Cape Town, before any southern excursions began, and again at Queen Charlotte's Sound in New Zealand, where the *Resolution* and *Adventure* rendezvoused. Any difference between real and calculated longitude during the time Furneaux and Cook were separated in the Indian Ocean appeared small when positions were calculated, and so this drift was ignored for this segment. However, over *Adventure's* final journey in late 1773-early 1774, from New Zealand across the Pacific and Atlantic to reach Cape Town, when the chronometer had aged by almost 2 years and no land was seen for some 3 months, the timepiece's reading had drifted so that the log's recorded longitude was ~ 17° out by the time *Adventure* reached Cape Town on 3 March 1774. Presumably, lunar observations had helped Furneaux identify his real longitude roughly, as some 10 days earlier he had changed course from tracking near the 50th parallel of latitude to head essentially due northwards towards Cape Town (Fig. 1). The data used here for this part of the *Adventure's* voyage have therefore had the longitude corrected assuming the chronometer slowed uniformly over the 81 days it took to sail from New Zealand to Cape Town. This only affects iceberg observations, as no sea ice was observed by the *Adventure* whenever it was separated from the *Resolution*.

For each day iceberg density as noted in the journals and logs was recorded, with '0' denoting no "islands of ice", '1' if one "island of ice" was noted, '2' if a few icebergs were seen, and '3' if an iceberg field was noted. For some observations an idea of the size of an iceberg is given, in terms of circumference or height – the presence of such a record is flagged in the spreadsheet (Bigg, 2024), although not used in this analysis. A flag is also noted in the dataset if any of the ships stopped to harvest iceberg fragments to supplement their drinking water.

Sea ice is also noted in the journals as 'loose ice', 'field ice', 'drift ice' or 'pack ice', clearly different from 'ice islands'. If no sea ice is mentioned a value of '0' is noted, but if 'loose ice' or 'drift ice' is present a value of '1' is given, with '2' when the sea ice observed is clearly more extensive.

**2.2 Modern sea ice data**

To compare the sea ice fields from those provided by the past documentary evidence with current observations, daily high resolution fields are required. Microwave brightness temperatures from 5 satellite instruments are available to give a daily coverage of sea ice concentration over northern and southern polar latitudes at a resolution of 25 km x 25 km since August 1987, with bi-daily data back to November 1978 (Parkinson et al., 1999). There has been intercalibration between changing

sensors over the years, and infill of errors (Cavalieri et al., 1999), meaning that this dataset is robust and well able to look at

climate trends and anomalies (Parkinson, 2019). Southern Hemisphere daily fields were extracted, for available years, for the days on which sea ice was found by Cook or Bellingshausen from the National Snow and Ice Data Center (https://nsidc.org/data/nsidc-0051/versions/2). This gave between 38 and 40 years of daily values, which were used to provide a statistical measure of the mean sea ice extent and its latitudinal extremes for the specific longitudes of the eighteenth and nineteenth century observations. These extracted daily values of the modern northern sea ice edge for each

date of Cook and Bellingshausen's sea ice observations are provided in a Supplementary Spreadsheet (seaiceobs_moderndailyextremes.xlsx). Note that a 25 km square was determined to contain sea ice if the measured concentration was $\geq 15\%$.

### 2.3 Modern iceberg data

There are several sources of modern iceberg data, which can be used to provide a climatological view of iceberg density and

prevalence across the Southern Ocean. The main source that is used here is the long-term small (< 3 km) iceberg distribution data provided by Tournadre et al. (2016) and updates in the altiberg database (https://cersat.ifremer.fr/fr/Data/Latest-products/Altiberg-a-database-for-small-icebergs). This was produced from data across 8 altimeters on board satellites over 1991-2019; here the merged product from across the altimeters is used (Tournadre et al., 2016; https://cersat.ifremer.fr/fr/Data/Latest-products/Altiberg-a-database-for-small-icebergs). Iceberg position, size and volume is

calculated from the Doppler return altimeter data; the summary variable giving the monthly probability of an iceberg occurring in a given 100 km square is used here (https://sextant.ifremer.fr/geonetwork/srv/api/records/695647ad-5af3-427f-afed-1485d3458b93). The data is available via ftp://ftp.ifremer.fr/ifremer/cersat/projects/altiberg/v2/ and the data manual is available also through ftp at ftp://ftp.ifremer.fr/ifremer/cersat/projects/altiberg/v2/documentation/ALTIBERG-rep_v2_1.pdf. The smaller number of larger icebergs (> 5 km) have been tracked using scatterometer data since 1992 (Budge and Long,

2018). This gives a measure of the full iceberg presence envelope, although Cook's largest observed iceberg probably was ~ 3km diameter (Martin et al., 2022), meaning the smaller iceberg distribution is a more appropriate comparator. There are also databases of iceberg observations compiled from historical data (Headland et al., 2023), shipboard observations from Russia (Romanov et al., 2017) and a combined Norwegian and Australian shipboard database (Orheim et al., 2023) which can be used to modify the view produced from the base altimeter iceberg database. All of the latter have records of location,

date and, in some cases numbers and sizes of icebergs; here only location is used.

### 3. Analysis and Results

### 3.1 Sea ice

Despite the circum-Antarctic nature of the voyages of both Cook and Bellingshausen, their sea ice records are both largely confined to the Atlantic and Pacific sections of their expeditions (Fig. 2). The sea ice records also occur across three (Cook)

or two (Bellingshausen) summers, as well as across at least two of the summer months as well. This makes comparison between and within them, as well as with modern day sea ice extent, non-trivial. The data will therefore be cross-compared according to the sea ice record for the specific days of the year when the voyages recorded sea ice. However, before this is examined, it is worth noting two points from Fig. 2. The first point is that in almost all areas pack ice and loose ice occur close together spatially, and between voyages. The second point is that there is a clear exception to this around 10-25ºE, where Cook's 1772/3 austral summer loose sea ice record is almost 10º further north than Bellingshausen's 1820 pack ice record. Indeed, the latter is essentially noting the ice adjoining the Antarctic continent itself off Queen Maud Land. However, modern sea ice extent is highly variable from year to year. For example, on 31st December, when Cook observed sea ice at 13.5ºE, 60.33ºS in 1772, the modern latitudinal variability over 1978-2022 of the edge of the 15% concentration band of sea ice at the same longitude of 13.5ºE has ranged over 60.19º – 69.91ºS (Fig. 3). In early summer there is often a tongue of less concentrated sea ice extending eastward at ~60ºS from the Antarctic Peninsula across the northern Weddell Sea. Note that polynyas can also occur within the Weddell Sea, as suggested in Fig. 3, meaning it should be remembered that the early explorers were being stopped by the first impenetrable ice barrier, rather than a continuous ice pack reaching to Antarctica.

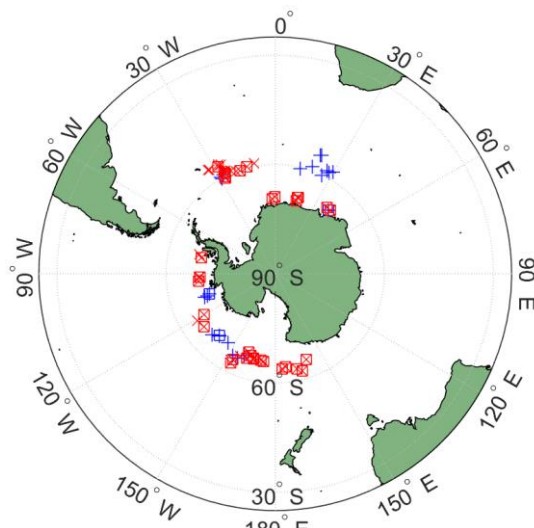

**Figure 2: Sea ice records from the voyages of Cook (shown by '+' in blue) and Bellingshausen (shown by 'x' in red. Pack ice (or landfast ice) is denoted by the voyage symbol enclosed in a square; loose ice is given by the voyage symbol alone.**

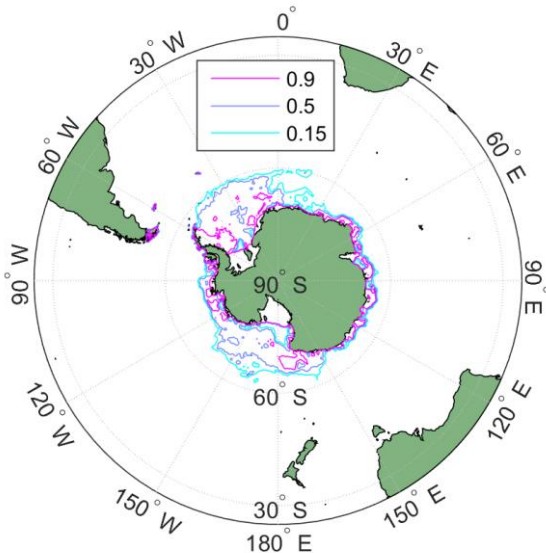

**Figure 3: An example of summer sea ice concentration in the Southern Ocean, from 31 December, 2007, showing areas of open water poleward of the outer sea ice edge. This paper uses a 15% concentration boundary to denote the ice edge, here given by the light blue contour. Note polynyas (regions of lower ice concentration) at the edge of the Filchner and Ross Ice Shelves.**

The statistical background to the sea ice records from Cook and Bellingshausen is shown in Fig. 4a, where all their records of sea ice, whether loose or pack ice, are shown by longitude, with a superimposed measure of the variability of the modern day northern sea ice limit for the same longitude ($\pm$ 0.15°). The latter is shown as a bar whose centre is at the mean latitude of sea ice observed for that day over the 38-40 values of the available daily microwave data from 1978-2022, with standard error bars denoting the variability. However, the extreme interannual variability of Southern Ocean sea ice extent means that it is also necessary to show on Fig. 4 the extreme northern sea ice edge over 1978-2022 for the given day and longitude as well, to capture the full potential variability in which to set the 18th-19th century data in proper context. It is worth noting here that Worby and Comiso (2004) found that satellite-derived ice edges tend to be 0.75$\pm$0.61° south of those observed in situ; this additional uncertainty in the modern data does not change the basic arguments of the discussion to follow. It is also worth noting that in general Cook's sea ice observations cam from earlier in the summer (Fig. 4b), when sea ice extent is intrinsically more variable as it moves towards the summer minimum at interannually variable rates (Parkinson et al., 1999). This increases the range of variability seen in Fig. 4a in many of Cook's observations compared to Bellingshausen's later summer data.

Fig. 4a shows that almost everywhere the Cook and Bellingshausen sea ice limits are well to the north of the modern standard deviation variability for microwave observations for the same day and longitude. However, in many longitudes these 18th-19th century values are still within the extreme envelope of modern observations over 1978-2022. Both Cook and Bellingshausen's exploration years were therefore extreme from an ice perspective, but usually not unprecedented in terms

of the modern, post-1978, record. Nevertheless, there are some regions where the historical observations are more northerly than today. Cook's 1772/3 sea ice records from 15-30ºE and Bellingshausen's 1821 records over 15-30ºW are some 2-3º

further north than the most northerly extreme of the satellite era (see centre of Fig. 4a (50ºW-50ºE)). Thus, the Weddell Sea ice tongue extended further east than during any year in the satellite era in the austral summer of 1772/3, and its Scotia Sea beginning was further north in 1820/1.

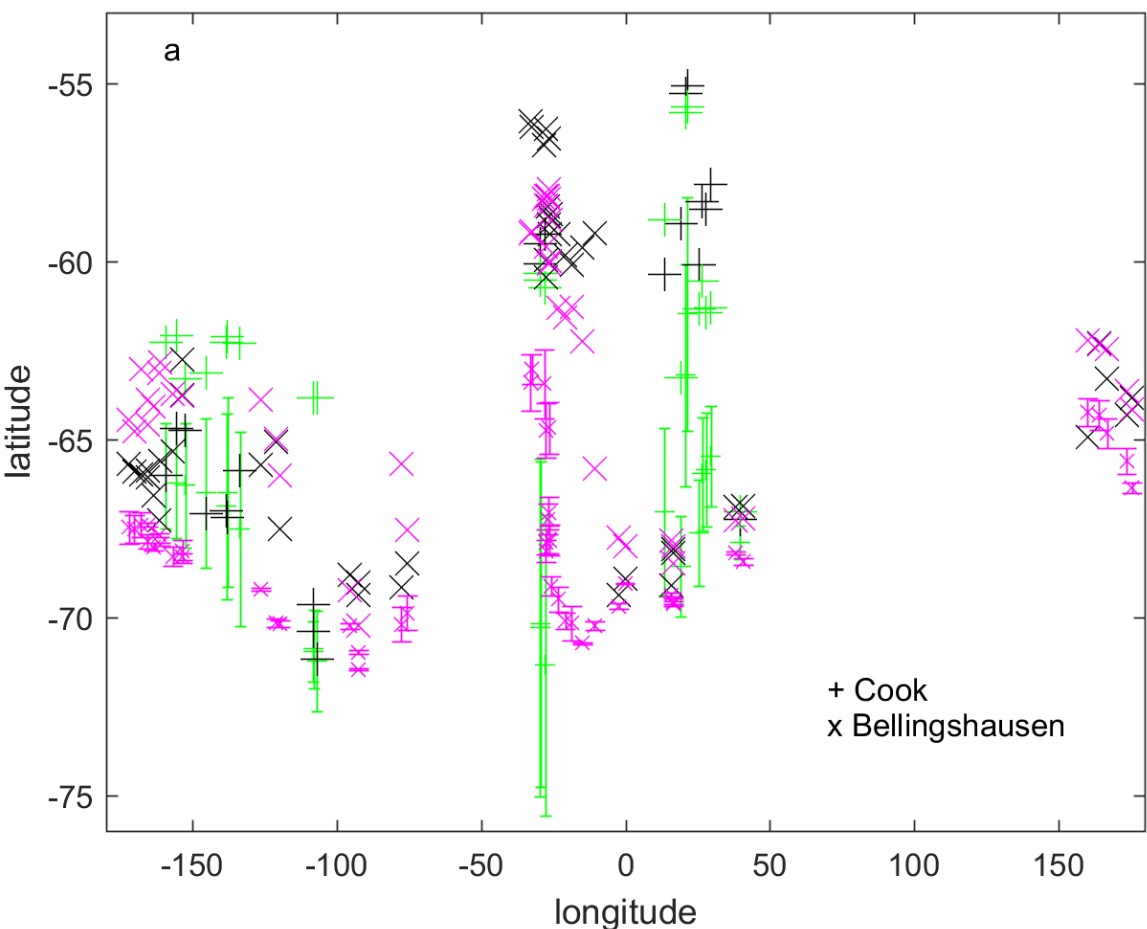

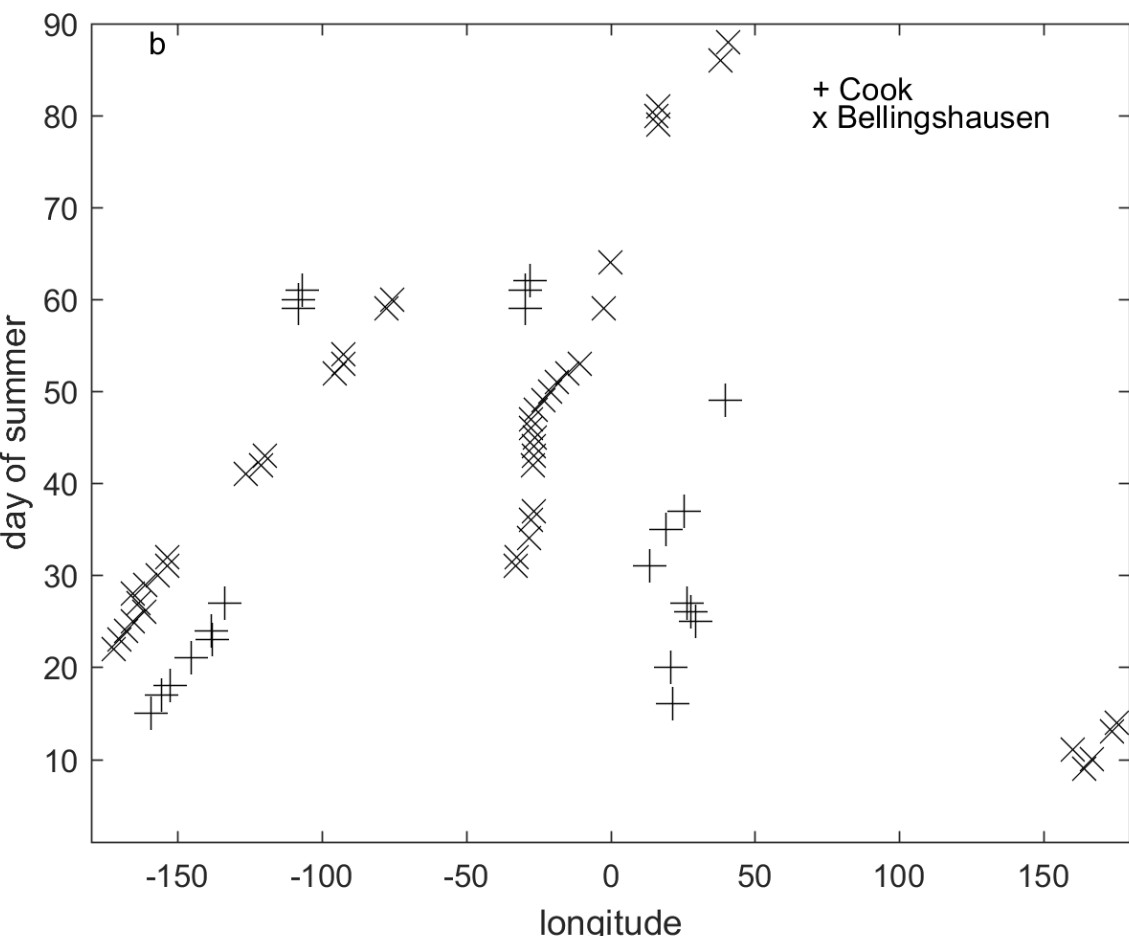

**Figure 4: a) Comparison of sea ice observations of Cook ('+') and Bellingshausen ('x'), marked in black, with the statistics of the 1978-2022 daily satellite record for the day of the year and position of each observation. The mean and standard error bars are shown for each modern satellite series, as well as the extreme northern modern record for each series. Modern data corresponding to Cook's records are shown in green, while Bellingshausen's are shown in magenta. b) Day of the summer, relative to 1 December, of the observations of Cook ('+') and Bellingshausen ('x'). Note that in general Cook's sea ice observations were from earlier in the summer than Bellingshausen, when sea ice is intrinsically more variable interannually.**

**3.2 Icebergs**

Icebergs were extensively recorded by Cook, Furneaux, Bellingshausen and Lazarev around the polar latitudes of the Southern Ocean (Fig. 5). Apart from 130-160ºE, where none of the expeditions sailed in sub-polar latitudes as they headed for temperate climes to over-winter, only the area of the sub-polar Atlantic to the west of the South Orkney Islands (~45ºW) has a minimal number of iceberg entries.

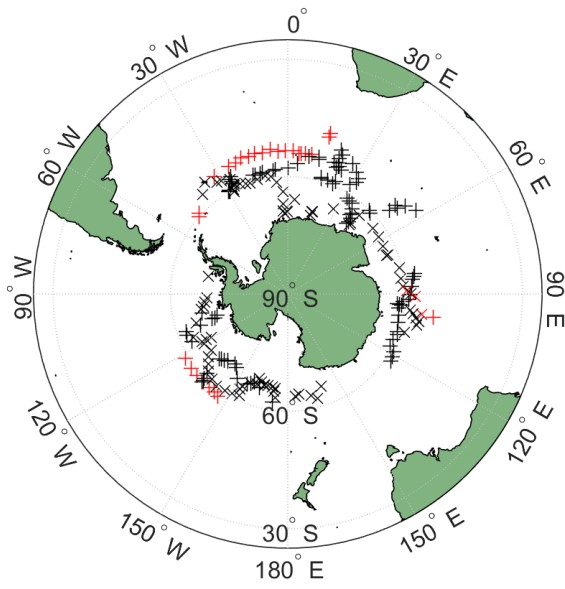

**Fig. 5: Iceberg observations recorded by the voyages of Cook ('+') and Bellingshausen ('x'). These include any iceberg sightings, so cover all records of category '1' to '3'. Note that additional observations from the *Adventure*, for Cook's expedition, and the *Mirny*, for Bellingshausen's, are shown in red.**

Icebergs were encountered on these 18th-19th century voyages largely in parts of the Southern Ocean where there are
altimeter records of icebergs in recent decades (Fig. 6a). The exception to this is in the Indian Ocean between 15-60°E, where in both 1773 (Cook) and 1774 (Furneaux) icebergs were encountered further north than current limits. This expansion of the Weddell iceberg tongue (see the high probability tongue in Fig. 6a) is consistent with other records from the late 18th century in this area (Martin et al., 2022). It is also notable that where Cook encountered extensive iceberg fields in the 1770s in the Atlantic is today mostly in regions of occasional iceberg encounters (Fig. 6b). Thus, both iceberg and sea ice records
are consistent with more icy South Atlantic and Indian Ocean sections of the Southern Ocean in the late 18th century.

In contrast, the Indian Ocean zone where the outlet from the Amery Ice Tongue is today was largely in the same position in the 18th and 19th centuries (Fig. 6a) and the presence of Pacific Ocean icebergs during the presently studied voyages matches today's records. Nevertheless, Bellingshausen in the early 19th century tended to encounter more icebergs north of the Ross Sea and West Antarctica than are found today (Fig. 6b), but still within the bounds of today's observations.

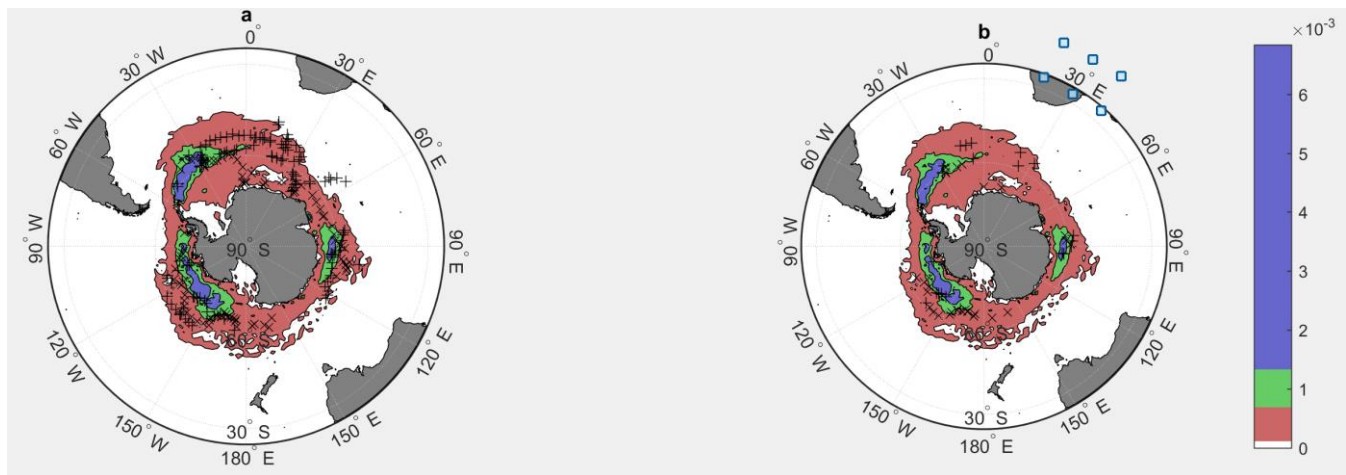

**Figure 6: Comparison of modern iceberg distribution from Tournadre et al. (2016) with iceberg observations of Cook ('+') and Bellingshausen ('x'). a) upper panel shows all iceberg sightings; b) lower panel only shows those iceberg sightings of category '3', namely iceberg fields. Units of modern distribution are mean probability of an iceberg being present in a 100 km square (see discussion in section 2.3).**

## 4. Discussion

The analysis section above compares the marine ice record from the Cook and Bellingshausen expeditions with modern day satellite-derived datasets, placing both the sea ice and iceberg records in the context of the most representative datasets available for the last few decades. However, there are other reconstructions of historical sea ice extent and recent iceberg distributions that it is worth comparing with the 18[th] and 19[th] century records presented here. There is also the recent independent analysis of Cook's iceberg record by Martin et al. (2022) with which it is possible to verify the current reconstruction. Here, these comparisons will be examined before concluding with a summary of the key findings from this analysis.

The first discussion of a subset of Cook and Bellingshausen's sea ice records was given by Parkinson (1990). She noted the greater sea ice concentrations seen by Cook in the Weddell Sea in 1772, consistent with the current data, and the similarity of both Cook and Bellingshausen's sea ice data in the Pacific sector, again consistent with present data.

Dalaiden et al. (2023) used a data assimilation approach using proxy sea ice records from land-based ice cores and tree rings around the Southern Ocean to reconstruct regional sea ice anomalies back to 1700. These regional reconstructions are similar, in general trends, although not in detail, with the satellite record and Fogt et al.'s (2022) proxy reconstructions of 20[th] century regional sea ice anomalies. This comparison is least successful in the Bellingshausen/Amundsen Sea sector, where the Dalaiden et al.'s reconstruction tends to overestimate the sea ice extent compared to Fogt et al. (2022) or modern observations. Most regions, however, suggest sea ice during the period 1750-1850 was somewhat more extensive than today (see Fig. 1 of Dalaiden et al., 2023), although the error bars tend to overlap with modern observations. The Weddell Sea is the region where the trend towards reducing sea ice in the last century was most pronounced. These results are consistent

with Cook and Bellingshausen's records of more extensive sea ice generally, but particularly in the Weddell Sea sector, that was shown in Fig. 4. This is also consistent with the finding of Love and Bigg (2023) that there was more extensive, and an eastward extension of, sea ice in the main summer months in the Weddell Sea in the 1820s-1840s.

For our iceberg comparison of today's record with Cook and Bellingshausen shown in Fig. 6 the satellite altimetric record of

310 Tournadre et al. (2016), updated to 2019 (https://cersat.ifremer.fr/fr/Data/Latest-products/Altiberg-a-database-for-small-icebergs), was employed. The latter is a dataset of all icebergs of area between ~ 0.1-8 km$^2$, which matches the range of iceberg sizes reported by all ships examined in this paper. However, there are also large databases of modern large icebergs > 5 km diameter, from scatterometer data (Budge et al., 2018), and modern ship observations (Romanov et al., 2017; Orheim et al., 2023). The larger iceberg dataset largely falls within that of Tournadre et al. (2016), shown on Fig. 6a. However, in the

315 case of both of the ship observation datasets, largely from independent sources Russian in the case of Romanov et al. (2017) and Norwegian Polar Institute and Australian sources for Orheim et al. (2023), their northern bounds of iceberg presence extend rather further north in many areas than the satellite-derived datasets. Note that the shipboard observations cover a significantly longer time period than the altimetric dataset of Tournadre et al. (2016). These shipboard observation limits are schematically overlain on the Cook and Bellingshausen voyages' distribution in Fig. 7. The vast majority of the 18th and 19th

century iceberg records lie within the 1950-2010 ship record, however, note that Cook's late January 1773 iceberg records from ~50ºE still lie outside the modern ship record. A few years later, in December 1789, Edward Riou encountered icebergs at ~ 44ºS, 45ºE (Martin, 2023), even further north than modern or Cook's records, if further west than the latter's extreme record. There were clearly extensive and unusual iceberg numbers in the South Atlantic and southwestern Indian Ocean during the 1770s and 1780s, perhaps indicating a recent period of calving of very large giant icebergs from Antarctica.

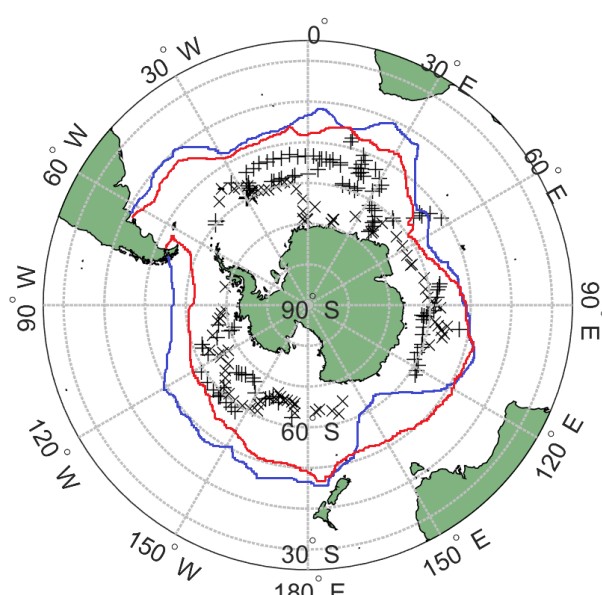

**Figure 7: Iceberg observations of Cook ('+') and Bellingshausen ('x') explorations in context of modern ship-board iceberg observations. These include records of all iceberg categories from '1' to '3'. The red line is the northern limit of iceberg observations from Romanov et al. (2017), while the blue line is the northern limit of iceberg observations from Orheim et al. (2023).**

The dataset of icebergs from Cook's voyage provided by Martin et al. (2022), and readily visible in Fig. 2 of their paper, very strongly corresponds with those given here from different sources for the same voyage. One does need to examine several sources for the same voyage where available: for example, there were a few occasions where Forster had noted icebergs for a particular day (Forster, 1981), but Cook's 1775 journal had not, and vice versa. The present data is also enhanced by icebergs records from the *Adventure's* period of independent sailing (Fig. 5), which was responsible for the increase in data in the current work for the South Atlantic and the southeastern Indian Ocean.

## 5. Conclusion

The Introduction of this paper ended by noting our principal hypothesis that it was expected that sea ice and iceberg records would generally extend further north than those today in both the two circum-Antarctic expeditions considered here, as they occurred at the end of the Little Ice Age. Our analysis has confirmed this hypothesis for both sea ice and iceberg records, with both showing more northerly limits than is typical for the current day (Fig. 4 and Fig. 6). Nevertheless, the Southern Ocean climate and ice record is very variable from year to year (Fig. 4), so in most areas the 18th and 19th century data falls within the most northerly limits of extreme years today. The exception to this lies in the South Atlantic where especially Cook's expedition experienced marine ice of both forms further north and east than is likely today, even in extreme years. While the iceberg anomalies experienced by Cook and Furneaux may have been due to extraordinary giant iceberg calving events from Antarctic ice shelves from previous years to decades, such events would have been extreme compared to the modern record. It is much more likely that the marine ice found by Cook and Bellingshausen reflect a colder than average climate of the South Atlantic in particular, and the Southern Ocean more generally (Dalaiden et al. 2023), during 1770-1820 than today.

**Data availability.** The full dataset of sea ice and iceberg observations from the source logbooks and journals is available through the UK Polar Data Centre at https://doi.org/10.5285/bcbc7d2e-4e75-43ad-8d9c-a3a3bd4fb013 (Bigg, 2024).

**Author contributions.** The document transcribing, statistical analysis, preparation of the diagrams and the writing of the text was carried out by GB.

**Competing interests.** The contact author has declared that they do not have any competing interests.

**Acknowledgements.** The author would like to thank Seelye Martin for previously alerting the author to his 2022 work on Cook.

**Financial Support.** The analysis for, as well as the preparation and publication of, this paper was funded by a Leverhulme Trust Emeritus Fellowship (grant no. EM-2022-042).

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
