# Peer review of "The Southern Ocean marine ice record of the early historical, circum-Antarctic voyages of Cook and Bellingshausen"

_Climate of the Past, 2024_

## Referee Comment (RC1)

Review of *The Southern Ocean marine ice record of the early historical, circum-Antarctic voyages of Cook and Bellingshausen* by Grant R. Bigg.

This is a well-written paper that suggests that 200-250 years ago, the ice in the Weddell Sea extended further north than at present. In its present form however, the paper has several issues that must be dealt with before publication.

Although the author cites the Claire Parkinson 1990 paper on the properties of Antarctic sea-ice cover during the same period in his reference list, there is no discussion or citation of her paper in his text. In Parkinson (1990), she used some of the same observations of Cook and Bellingshausen that the author uses in the present paper. She also used observations by Charles Wilkes in 1838–42, and James Clark Ross in 1839–43, which the author does not use, and appears to be the first person to discuss Cook's Weddell-Sea ice tongue.

Parkinson concludes (her abstract) that "When these locations are compared with satellite-derived ice edge locations in the mid 1970s, there is a suggestion of particularly heavy ice covers in the eastern Weddell Sea in December 1772, in the Amundsen Sea in March 1839, and perhaps, on the basis of an isolated observation, in a portion of the western Weddell Sea in January 1820." These results are sufficiently similar to the author's that they need to be described and commented on. The author then needs to show how his improved data set expands on this work.

Another issue concerns the supplementary data, which are embargoed and thus unavailable to this reviewer. These data must be made available to reviewers before acceptance. Further, please add Bigg (2024) to your reference list.

In the embargoed data set, Lines 139-140 state that the author has recorded iceberg density, as in "For each day iceberg density as noted in the journals and logs was recorded, with '0' denoting no "islands of ice", '1' if 1 "island of ice" was noted, '2' if a few icebergs were seen, and '3' if an iceberg field was noted. [note that in this sentence, the author should write 'one' island of ice, not '1.'] However, you don't use this information in Figure 5, and use it without reference to your '1, 2, 3' system in Figure 6. Why not? Please be clear about how you use your data set, on which you have expended a great deal of work.

My feeling is that given the greatly expanded data set which the author developed, that once the problems with the Parkinson reference and work are corrected, his data sets made available to the reviewers, and certain issues about the figures given below are corrected, the paper should be suitable for publication.

Sincerely,

Seelye Martin

**Specific concerns:**

The Bellingshausen and Cook cruises are almost 50 years apart. Bellingshausen's cruise was part of the 1820 discovery of the Antarctic Peninsula (him, Palmer, Bransfield...), and also about the same time as Weddell's 1823 record setting voyage into the Weddell Sea. Could you use Weddell's data as well?

I applaud your use of Forster's data, and especially of Tobias Furneaux's data, but what about William Wales, the astronomer on the *Endeavor*? From my experience, Wales did a great job, but unfortunately, his journal is only available in handwritten form. Somebody, someday, has got to go through this for the icebergs.

Line 32: Although interesting, your comment about Arctic exploration and Horatio Nelson is not relevant to the argument of your paper.

Line 86-88: what about in the 1840's, when James Ross, Jules Dumont d'Urville, and Charles Wilkes (United States Exploring Expedition of 1838–1842) did their coordinated surveys? Ross, and maybe the others, were part of an international effort to carry out von Humboldt's coordinated study of the Earth's magnetic field.

Line 69, "New Style, see section 2.x for discussion of dating:" Define 'x'.

General question: Are your colors accessible to people who suffer from color-blindness?

Line 115 ff: good discussion of chronometer accuracy.

Line 143: Change "clearly different to 'ice islands'", to "clearly different *from* ice islands."

Line 208, Comiso/Worby comment: Does their systematic error affect your results?

**Concerns about Figures**

Figure 1: There is a problem with the fish-hook in the green line south of Cape Town, why does this line terminate abruptly?

Figure 2: I find your combination of the colors and symbols confusing, especially since you don't define the colors in either your caption or on the figure. Why not put all your symbol information in your caption?

Figure 3: "An example of summer sea ice concentration in the Southern Ocean, from 31 December 2007." Any particular reason for this choice of date? Am I supposed to compare this figure with Figure 2 above?

Figure 4: In concept, this is a good figure, but in practice, it is extraordinarily difficult to understand. Have you thought about doing it as a polar projection? The mixes of color and symbols here makes it hard to read. For example, the small black x's are defined as Bellingshausen data, but there are no such x's. The black x's are large, and the small x's are purple. Why? You should explain all the colors and symbols in the caption. Further, the light green slightly florescent color tends to fade into the background, can you find a better color? Your data bleeds over the top of the figure, so please extend the figure north. Note that the horizontal and vertical axes should be labeled 'degrees latitude and longitude.' Can you flag the Parkinson ice-tongue anomaly? Is it within bounds? You say in the text (line 216) that it's an east-west anomaly, but I sure don't see it.

Figure 5. Iceberg observations: You show the crosses and the x's for Cook and Bellingshausen on the figure but define the colors in the caption. I'd feel better if you'd either put all explanation in the legend on the figure or in caption. Also, since you have done a great deal of iceberg analysis and sorting into the '1, 2, 3" categories, why not use this information on the figure?

Figure 6. Comparison of modern iceberg distribution from Tournadre et al. (2016) with iceberg observations of Cook and Bellingshausen: Instead of referring to the top panel and bottom panels, why not call these Figures 6a and 6b? Top panel shows all Cook and Bellingshausen iceberg observations; lower panel is where iceberg fields were. I assume that your upper figure shows all of your icebergs, while the lower figure shows your iceberg fields, namely your Category '3' observations. Is this correct?

Figure 7. Comparison of iceberg observations from Cook and Bellingshausen explorations with modern ship-board iceberg observations from Romanov et al. (2017) and Orheim et al. (2023): Need to explain the symbols and need to define crosses and x's. Pattern in Weddell Sea/South Atlantic appears to be shifted east, comments? An interesting thing about this figure is that in the Weddell Sea/South Atlantic, the icebergs and sea ice appears to be shifted east; is this correct?

END

---

## Referee Comment (RC2)

**Review of the manuscript by Grant R. Bigg (2024) for the Climate of the Past**

**The Southern Ocean marine ice record of the early historical, circum-Antarctic voyages of Cook and Bellingshausen**

In general, I find that the manuscript is well written but at times lacks detail and consistency. I believe that if the author addresses concerns of reviewer 1 (most of which I am echoing but not repeating here since I will include this analysis in the editor decision) and those listed below, the manuscript would be ready for publication. Below is the list of my major and minor concerns that I would like to be addressed in a revised version of the manuscript prior to publication.

**Major concerns:**

The introduction section 1 is very long but there is no overview of the state-of-the art for the second major aspect of the study – analysis of modern techniques for sea ice and iceberg identification and mapping that are central to this research. It would also be helpful to create a table with journey chronologies, including locations and ship names, among others. This will save a lot of space on the detailed description of these journeys and their parameters. Currently the description is quite tedious to read.

Sections 2.2 and 2.3: Details are painfully missing. Readers need more explanations for how the data has been derived and from what observations. We need to ensure that readers do not need to read several articles to be able to understand what has been done in this one. Please, expand and clarify.

The beginning of Section 3: I feel that the first paragraph rather belongs to the introduction. It can be condensed to a signle phrase in the opening of Section 3 and refer the reader to the introduction where the overlap and apparent differences with Martin et al. (2022) are discussed.

Figure 4: Why are the uncertainties so large for the modern analogues of Cook's data and so small for those corresponding to Bellingshausen's data? Some discussion of this phenomenon would be helpful.

**Minor concerns:**

I am not a native speaker but would not „temperate climate" be better in this case than „temperature climes" in line 53 and in similar instances?

Line 79: but again WERE prevented

Figures 1-3 and 5-7: Please, correct the longitude placement for the interval of 90W to 90E. Currently they have half-sunk into the frame.

Figure 2: Increase the size of the figure (task for the typesetting). It is impossible to see the details of the figure in the current format.

Figure 6: From this article I do not completely understand how these probabilities are calculated. During a certain period? I could go to the source of the data but the point is that this information should be provided as independent in the current study. Again, a reader should not be forced to consume other articles to understand this one. Please, also explain the scale better and discuss these results in a greater detail.

Line 269: Move comma after the reference.

---

## Author Response (AR1)

Point-by-point response

***Reply to comments from Seelye Martin***

**I have added suitable comments in the Introduction and the Discussion sections looking back to the Parkinson reference to extensive sea-ice in the Weddell Sea.**

**I have had the UKPDC remove the embargo in anticipation of the paper being accepted. The supplementary dataset is now freely available via Bigg (2024) in the reference list.**

**With respect to the iceberg part of the dataset I have amended the text in section 2.3, as suggested by Dr. Martin. I have expanded the Figure 6 legend to make it clear part b uses the iceberg field data (ie. Category '3').** In Figure 5, I decided for clarity in an already busy figure to only note the presence of icebergs. This is the summary iceberg figure, and the detail of iceberg density is better left to Figure 6, for comparison with the modern observations. I stand by this division, which I think makes the paper clearer.

Specific Concerns

**I have said more about the 1820s record in the Discussion and I have added an additional couple of sentences in the revision on what Love and Bigg found of relevance here.**

I agree with Dr. Martin that Wales' journal of Cook's expedition needs to be checked for the iceberg data. It is worth noting however that while there are a few occasions where Forster recorded icebergs and Cook's journal did not, on the whole the two records were very similar so I suspect that Wales' journal is unlikely to add too much.

l. 32: I think it is an interesting context that a British expedition was seeking the North Pole at the same time as Cook's expedition south, but that they seem not to be linked. **While of tangential importance to the paper overall I believe it is an important contextual observation regarding contemporary polar expeditions and do not wish to remove it. The editor has agreed with me retaining this.**

l. 86-88: **I have expanded the section in the Introduction on later nineteenth century expeditions, while keeping it clear that Cook and Bellingshausen offer unique, circum-Antarctic perspectives worthy of a paper in its own right.**

l. 69: apologies. **This should be section 2.1. I have corrected this in the revision.**

l. 143: **This text has been changed as suggested in the revision.**

l. 208: The Comiso and Worby result doesn't have a significant impact on the analysis (note the large error bar). I already note this in the second phrase of the sentence.

Figure 1: The fish-hook south of Africa is an illusion. The outbound green line starts only a few days before ice was observed, and so is well south of Africa at the top if the fish-hook, while the inbound green line goes all the way to Cape Town. **I have added a note to the Figure 1 legend explaining this, as well as information about the exploration voyage section's in a new Table 1.**

Figure 2: I agree this Figure could be confusing. **I have put all the colour (Cook is red Xs, Bellingshausen is blue crosses) and shape information into the Figure legend in the revision.**

Figure 3: The purpose of this figure is to demonstrate that there may be open water or polynyas south of the northern sea ice limit, particularly, but not exclusively, in the Weddell Sea. I tried to find

an example where this was most obvious, and related to one of the observation days from both Cook and Bellingshausen's journals. The text does imply this, but is not too clear, so **I have added some explanatory text to the Figure legend in the revision.**

Figure 4: I welcome the helpful comments regarding this pivotal figure. **In the revision I have extended the y-axis northwards and southwards, to make all points clearer. I have altered the size of the symbols so that the historical and modern observations are the same size. I have made clear the Weddell ice tongue area in the textual discussion accompanying this figure. I have added a panel b to this figure, in response to the second reviewer's comments, showing the day of the summer for each historical observation, to help explain the variation in sea ice extent variability around the Antarctic.**

Figure 5: **I have put all the information about the figure in the legend in the revision. This is a summary iceberg figure – I use the extreme iceberg field data in the lower panel of Figure 6 and suggest this is a clearer way of allowing the reader to see the difference.**

Figure 6: **I have changed the panels to be Figures 6a and b, as suggested, both in the Figure and references in the text, in the revision. I have add a note to the legend to explain that the bottom panel is indeed showing the category '3' iceberg data.**

Figure 7: **I have put all the figure colour and symbol information in the legend, rather than mix between legends on and off figure, in the revision. The iceberg data is the same as in Figure 5, but it is true that some ship observations go further east than the Tournadre data of Figure 6 suggests. I think this is because the ship data goes further back in time and have added such an explanatory note to the text discussion in the revision. I do currently note in the Conclusion that marine ice (ie sea-ice and icebergs) suggest a colder Weddell Sea in the past than today, but have made it more explicit in the revision that it is both icebergs and sea-ice that show this eastwards trend.**

***Reply to comments from Irina Rogozhina***

Major Concerns

The Introduction section was difficult to write as it needs to combine background with some discussion of the two expeditions' timeframes and comparisons. **The suggestion of adding a Table to make the timings of the different ship voyages clearer is an excellent one. I have added this Table in the revision and made more explicit reference to it, and Figure1, in the Introduction. I think it makes the surrounding text more directed and fitting to the narrative.**

I thank Dr. Rogozhina for pointing out that the modern data methods are not given sufficient background. I agree with her, on re-reading the paper. I think there is some risk of duplicating material if too much is added to the Introduction, while Sections 2.2 and 2.3 are improved by adding more detail. **I have enhanced both Sections 2.2 and 2.3 significantly, introducing new references in the modern sea ice methodology especially.**

I agree that much of the first paragraph of section 3 belongs in the Introduction. I have **moved much of this section 3 paragraph to expand the Introduction mention of the Martin et al. work. With this being done there seemed no need for an introductory paragraph to section 3.**

The apparent difference in uncertainties, or current variability, between the Cook and Bellingshausen data is almost certainly linked to the day of the year when a given ship was in a particular longitude. Cook spent less time in southerly latitudes, and this was mostly in the early summer months of December and January, while Bellingshausen was mostly in such latitudes in

January and February, when there is less sea-ice typically. **I have added a new panel for Figure 4 (4b) which shows the date in the summer for each historical sea ice observation. This has some new text in the accompanying text on the figure to explore the above points more.**

Minor Concerns

l. 53: **You are correct – I should have said "temperate climate" and have done so in the revision.**

l. 79: **Again, you are correct. I have added a "were" in the revision.**

Figs. 1-3 and 5-7: Thank you pointing out the offset of some of the longitude markers. This must be something to do with the matlab code **and I have corrected it in the revision. Please note that in doing this for Figure 7 the two modern ship-board iceberg limits needed to be redrawn by hand. This is an imprecise process and you will notice slight differences from the lines in the original figure. They are not significant and have not affected the discussion.**

Figure 2: The tif file for Figure 2 is the same physical size as for Figure 3. I think the small size in the pdf is an artefact of trying to fit the figure in the page in WORD. **I have provided the tif files for all figures directly in the revision.**

Figure 6: **I have made it clear in the revised section 2.3 how the iceberg probabilities are calculated by Tournadre et al. I have slightly expanded the discussion of Figure 6 in the text to make clearer the correspondence between past observations and modern "iceberg alleys" such as the Weddell iceberg tongue.**

l. 269: Thanks very much for pointing this out. **I have moved the comma to after the Budge et al. reference in a revision.**

---

## Author Response (AR2)

Reply to editor's comments 080824

**All my responses are visible in the track changes document cook_bell_rev2_trackchanges.docx that can be provided on request.**

I have gone through your revised version of the manuscript and find that it still requires some relatively minor changes and improvements. There are some typos here and there that should be corrected: For example, "cam" to "came" or "come" in line 238, and "includes" to "include" in line 267. The citation in line 297 has a wrong year. I am also unsure regarding the grammar in line 100; specifically, I am confused by the placement of "are".

**I have made all these corrections and amendments.**

Table 1: Maybe you could also include a column with the longitude ranges corresponding to each part of the expeditions. This will make Table 1 more meaningful, providing insights into sectors of the oceans that were covered at each stage.

**I hve converted the "sector" column into a more specific listing of the longitudes covered by each sub-section.**

Please, increase sizes of all circumpolar figures.

**This has been done in the WORD document. All tif files will be uploaded so the appropriate size can be created by the journal publication team.**

Figure 4b: Please, adjust the format to fit the style of Figure 4a.

**Lines have been added to the top and right of Fig. 4b.**

Figure 6: Please, place subfigures side by side. Right now, it is a very inefficient way to use journal space.

**Fig. 6 has been altered to go across the page rather than down.**

Sections 2.2 and 2.3: Regarding your response to my review, I would not call one extra sentence in Section 2.2 and two extra sentences in Section 2.3 a significant enhancement. More could be done to address the lack of more detailed information regarding these datasets and the analysis behind them. One could provide a table in the appendix where different satellite data and their specifics could be compared. From my point of view, "a range of satellite instruments" sounds very vague.
**I have amended the text in both sections 2.2 and 2.3 further. The editor should note that I am using datasets provided by other authors of reputable datacentres and have not done the inter-satellite comparison that she seems to believe I have done. The merged data forms part of the supplied datasets. I have made this clearer and have added more links to underlying documents provided by the data centres. I have added the daily series that underlie Figure 4 as a Supplementary spreadsheet for full disclosure. Personally, I don't think this is needed in the paper but it is now available and noted in the text if required.**

I also think that the discussion of differences in data uncertainties (Cook versus Bellingshausen) belongs to the main text instead of the figure caption.

This text is now effectively added to the main text, although the relevant word remin in the Figure legend for better explanation of the figure.